# Association between Food or Nutrients and Gut Microbiota in Healthy and Helminth-Infected Women of Reproductive Age from Zanzibar, Tanzania

**DOI:** 10.3390/nu16091266

**Published:** 2024-04-24

**Authors:** Aristide Toussaint Nguélé, Chiara Carrara, Matteo Mozzicafreddo, Hongliang Chen, Angela Piersanti, Salum Seif Salum, Said M. Ali, Cristina Miceli

**Affiliations:** 1School of Biosciences and Veterinary Medicine, University of Camerino, 62032 Camerino, Italy; aristidetoussaint.nguele@unicam.it (A.T.N.); chiara.carrara@studenti.unicam.it (C.C.); hongliang.chen@jlau.edu.cn (H.C.); angela.piersanti@unicam.it (A.P.); 2Department of Clinical and Molecular Sciences, Marche Polytechnic University, 60126 Ancona, Italy; m.mozzicafreddo@staff.univpm.it; 3College of Veterinary Medicine, Jilin Provincial Engineering Research Center of Animal Probiotics, Key Laboratory of Animal Production and Product Quality Safety of Ministry of Education, Jilin Agricultural University, Changchun 130118, China; 4School of Health and Medical Sciences, State University of Zanzibar, Zanzibar 146, Tanzania; salum.salum@suza.ac.tz; 5Public Health Laboratory Ivo de Carneri, Chake Chake 122, Tanzania; said@phlidc.org

**Keywords:** nutrients, diet, vitamins, minerals, food and gut microbiota correlations, helminths, *Ascaris*, *Trichuris*

## Abstract

Modulating the gut microbiota is recognised as one strategy for preventing and fighting diseases. While the significant impact of diet on the gut microbiota’s composition and function has been extensively researched, there is a notable lack of studies on the interactions between diet, microbiota, and helminth infections. Here, we used a combination of self-reported food intake and a 16S rDNA sequencing approach to analyse the composition of the gut microbiota in women of reproductive age from the two main islands of the Zanzibar archipelago, where helminth infections are endemic. We also applied a Spearman correlation analysis to food/nutrients and gut microbiota. Our results reveal that, despite close ethnic and cultural ties, the participants’ gut microbiota differs depending on their location. A nutrient intake analysis revealed deficiencies in minerals and vitamins, indicating an imbalanced diet. A correlation analysis identified bacterial taxa consistently correlated with specific food or nutrients in healthy women from both locations, and in two types of helminth infections. *Escherichia*/*Shigella* abundances, usually associated with *Trichuris trichiura* infection, consistently correlated with insufficient levels of vitamins B2 and B12. In conclusion, our findings suggest that the increased consumption of specific food like cassava and fish, as well as essential nutrients such as calcium, B vitamins, and vitamin A, may modulate the gut microbiota of populations residing in regions where helminth infections are endemic.

## 1. Introduction

The composition and role of the gut microbiota are widely acknowledged to have a significant impact on host characteristics, including metabolism, immune function, and defence against infections. Therefore, disturbances in the balance of gut microbiota and the subsequent disruption of host–microbiota interactions are known to be implicated in the development of numerous intestinal and non-intestinal diseases [1]. Gut bacteria also play a crucial role in supporting the host’s well-being by enhancing their intestinal defence mechanisms and aiding in the maintenance of normal gut functions. They provide various benefits for the host, including the regulation of gut movement; the synthesis of essential vitamins; the conversion of bile acids and steroids; the metabolism of foreign substances; the absorption of minerals; and the activation or neutralization of toxins, genotoxins, and mutagens [2]. The gut microbiota also plays an important role in energy harvesting. Thus, Bacteroidetes are recognized to have a positive correlation with the reduction in body fat, whereas the positive correlation between Firmicutes and obesity can be associated with a greater energy harvest [3]. Hence, the gut microbiota is associated with obesity and other nutrition-related diseases [4], and associations between diet and gut microbiota are being widely researched since diet is recognized to be a key determinant in shaping the composition and function of the gut microbiota [5]. According to De Filippis et al. [6], different individuals may have distinctive metabolic responses to the same food, depending on the microorganisms that they harbour. For some individuals, it was found that an higher relative abundance of one bacterium with respect to another, associated with a fibre-rich diet, produced weight loss, whereas obese individuals with higher abundances of certain microbes, treated with a hypocaloric, high-protein, and high-fibre diet, showed better metabolic outcomes (lower insulin resistance and LDL cholesterol) compared to those with a lower baseline concentration of the same microbes [6].

Additional evidence suggests that germ-free rats harvest less energy from a polysaccharide-rich diet and a reduced adiposity, despite an increased intake of food in comparison to their colonized counterparts [7]. Thus, the gut microbiota is an important environmental factor that affects energy harvest from the diet and energy storage in the host [8]. Those findings can justify why correlations between diet and microbiota are widely investigated. Interactions between microbes and vegan, vegetarian, and omnivore diets have been intensely explored [9,10]. Similarly, the Mediterranean diet has been linked to a healthy gut microbiota and changes in circulating metabolites [11]. Studies have shown that a strict adherence to the Mediterranean diet beneficially impacts the gut microbiota and associated metabolome [12]. Some bacterial taxa have thus been associated with specific nutrients, groups of foods, or dietary patterns. It was proven that individuals mainly obtaining nutrients from plant-food sources tended to have a more diverse gut microbiota than individuals obtaining nutrients from animal-food sources [13,14]. Some bacteria have been found to be associated with a high consumption of carbohydrates and with both vegetarian and vegan diets. Conversely, other bacteria belonging to the same phylum have been found to be associated with a high consumption of animal protein, amino acids, and saturated fats [15]. Despite the widespread adoption of this type of analysis, the presence of unusual [16] or inconsistent associations between some taxa and diets makes it difficult to draw conclusions. Moreover, while many studies have explored the associations between diet and the microbiota in the context of obesity [17] or inflammation [18], few investigations have been conducted in the context of infections with soil-transmitted helminths (STHs) [19]. STHs infect millions of people in subtropical and tropical countries. The infection particularly affects children, resulting in malnutrition, growth stunting, intellectual disability, cognitive deficits, and other manifestations [20]. In some endemic African regions, such as the Zanzibar archipelago (Tanzania), preventive chemotherapy has been conducted for many years. However, this approach is not effective enough to control STH morbidity [21]. Therefore, finding complementary strategies in the fight against helminth infection is an urgent priority. One approach could be the modulation of the gut microbiota since a relationship between gut microbiota and unsuccessful helminth therapies has been found [22,23,24]. 

In developing countries, women play a pivotal role in sustaining and nurturing the family unit by procuring and preparing food, as well as providing care to dependent family members, especially children [25]. Additionally, the initial colonization of bacteria in the gastrointestinal tract of naturally born infants is primarily attributed to the mother through the faecal–oral route [26]. Recognizing the significance of these factors, we argue that a fundamental step in the development of complementary strategies to limit malnutrition in children and fight against helminth infection is understanding how to improve diet and gut microbiota in women of reproductive age. 

Consequently, this study aimed to investigate the correlations between diet and gut microbiota in both healthy and helminth-infected women of reproductive age. The research was conducted among women residing in Pemba and Unguja, the two main islands of Zanzibar, where child stunting and malnutrition is being monitored by the World Health Organization (WHO). The overarching goal was to identify specific foods or nutrients that may promote the growth of beneficial bacteria or limit the proliferation of harmful bacteria.

## 2. Materials and Methods

### 2.1. Ethic Statements

This study was approved and authorized by the Zanzibar Health Research Ethical Committee (ZAHREC/03/REC/MARCH/2022/16). After an explanation provided in Swahili with the help of a nurse and a local community worker, all participants accepted and signed an informed consent before enrolment in the study.

### 2.2. Study Design and Recruitment of Participants

This cross-sectional study was conducted in the Zanzibar archipelago (Tanzania), where helminth infection and malnutrition are endemic. In total, 75 women of reproductive age (WRA) were recruited from the islands of Pemba (58 participants) and Unguja (17 participants). 

All participants were between 18 and 45 years old, had not taken antibiotics or probiotics within the previous two months, and had no symptoms of disease. A detailed questionnaire was filled in by the WRA and used for collecting information about their lifestyle, family, health, and nutritional condition. Finally, nutritional-related anthropometric parameters were measured. Participants meeting the inclusion criteria were provided with stool containers to collect stool samples.

### 2.3. Faecal Sample Collection and Parasitological Analysis

We strongly recommended that participants provide stool samples to the health facilities as soon as possible after emission. The contact with the participants was maintained by local nurses. In both islands, we worked with well-equipped research institutions, specifically the Public Health Laboratory Ivo-de-Carneri (PHL) in Pemba, and the State University of Zanzibar (SUZA) in Unguja. In Pemba, the sample collection was in sanitary centres (equipped with blue ice containers) located directly in the villages where participants were recruited. In Unguja, recruited participants were hosted in the Mnazi Moja Hospital for the regular sanitary checking of their children, and their samples were immediately stored at −20 by the nurses. After collection, stool samples were sent to PHL-IDC in Pemba and to SUZA in Unguja for parasitological analysis. There, each sample was divided in aliquots and stored at −20 °C before shipment to the University of Camerino, Italy, for DNA studies. The Mini–FLOTAC technique was utilized for microscopic examination. Briefly, two grams of stool samples were sufficiently homogenized with the flotation solution (saturated sodium chloride). After homogenization, the samples were added to the two flotation chambers. Finally, after waiting for 10 min, the number of eggs per gram of faeces was determined under a microscope. Analytic sensitivity allowed up to ten eggs to be identified per gram of faeces. This analysis was repeated twice for each sample by two well-trained laboratory technicians. The shape of the eggs is different and peculiar to each helminth species. Therefore, it was possible to detect the presence of *Ascaris lumbricoides* eggs and *Trichuris trichiura* eggs, or the concurrent presence of eggs from both parasites. After parasitological analysis, the results were delivered to the enrolled female participants to ensure that they could go to the sanitary centre to receive anti-helminth treatment, where necessary. 

The sample collection was all performed in about one month, specifically in May. 

### 2.4. Recording and Evaluation of Food and Nutrient Intake

All participants were asked to record their food consumption for one full week using a seven-day nutritional table provided by the research team. Then, the Tanzanian food composition tables prepared by the Harvard School of Public Health (Boston, MA, USA) in collaboration with the Tanzanian Ministry of Health were used to evaluate the average intake of food and nutrients [27]. The WHO guidelines for the nutrient profiles of African women [28] were used to evaluate the nutrient profiles of the participants. When WHO information was insufficient, we used EFSA recommendations for the nutrient intakes of women [29]. 

Although the structure of the diet was studied in all 75 participants, as well as the microbiota diversity and taxonomy, we excluded pregnant (12 WRA), obese (11 WRA), undernourished (3 WRA), and helminth-infected women (17 WRA) for the analysis of food/nutrient–microbiota correlations, to avoid specific conditions that could have an influence on the relationship between food and gut microbiota. One participant from which we could not get complete anthropometric parameters was also excluded from this analysis. The remaining were divided into healthy women from Pemba (18 WRA) and healthy women from Unguja (13 WRA). In addition, to detect the correlations between food and gut microbiota in the presence of helminth infections, we considered 8 WRA infected by *Ascaris lumbricoides* and 8 WRA infected by *Trichuris trichiura* (all from Pemba).

### 2.5. DNA Extraction, PCR, and Sequencing

DNA was extracted using the QIAamp Fast DNA Stool Mini Kit of QIAGEN. The DNA concentration and absorbance of each sample were evaluated using the NanoDrop™ One/One C Microvolume UV-Vis Spectrophotometer, Thermo Fisher Scientific, Waltham, MA, USA. Before sending the samples for sequencing, all extracted DNA was amplified through the conventional PCR. The primers used for this were the following:Pro 341F:5′-TCGTCGGCAGCGTCAGATGTGTATAAGAGACAGCCTACGGGNBGCASCAG-3′Pro 805R:5′-GTCTCGTGGGCTCGGAGATGTGTATAAGAGACAGGACTACNVGGGTATCTAATCC-3′

The PCR-amplified products were run at 120–124 volts for 24 min using gel electrophoresis, and then checked in a UV light room. 

Next, 50 ng of purified DNA from each sample was prepared and sent to the BMR Genomics company (Padova, Italy) for sequencing. Sequencing libraries were generated using the NEBNext® Ultra™ DNA Library Prep Kit (New England Biolabs, Ipswich, MA, USA) following the manufacturer’s recommendations. Library quality was assessed and sequenced on an Illumina MiSeq PE300 platform (Illumina, San Diego, CA, USA).

### 2.6. Bioinformatic and Data Analysis

The software QIIME2 (Quantitative Insights into Microbial Ecology, version 2023.5) was used to analyse 16S rDNA gene sequences generated using NGS technologies. Briefly, after filtering out low-quality reads (minimum quality score of 25, minimum/maximum length of 200–250, no ambiguous bases allowed, no mismatches allowed in the primer sequence, and no phiX reads/chimeric sequences), all remaining sequences were subsequently clustered into Operational Taxonomic Units (OTUs) on the basis of similarity following the DADA2 pipeline and using the QIIME 2 Plugin ‘*dada2*’ version 2023.5.0. Samples were evaluated for alpha diversity (microbial diversity within samples) and beta diversity (community diversity divergence between samples) calculations in QIIME2. We assessed the statistical significance of alpha diversity metrics using a two-sample *t*-test and a Kruskal–Wallis test as implemented in QIIME2. Taxonomic analysis was performed by matching OTU sequences with both the Silva and Greengenes databases. The raw reads were deposited into the NCBI Sequence Read Archive database (SRA accession number: SRP495566, BioProject accession number: PRJNA1088637).

### 2.7. Statistical Analysis

A differential analysis of taxonomy between two groups was performed using STAMP (Statistical Analysis of Metagenomic and other Profiles) software with Welch’s *t*-test (version 2.1.3) and two-sided 95% confidence interval. *p*-values < 0.05 were considered significant. From each island, WRA were categorised as either healthy or infected by helminths following microscopical examination. The composition of microbiota was analysed based on the location. The associations between food/nutrient intakes and gut microbiota were explored using Spearman’s correlation analysis, implemented through the ‘*cor*’ function of the R system. Then, we converted the correlation matrix into an adjacency matrix, setting a cutoff of 0.3 for both positive and negative correlations by comparing the absolute values. Spearman correlation heatmaps were generated using the R package ggplot2 (https://ggplot2.tidyverse.org/ (accessed on 26 August 2023)). The software GraphPad prism 9.1.5 was used to compare food intake between women from Pemba and Unguja. A Student’s *t*-test was also used and a *p*-value < 0.05 was considered significant. 

## 3. Results

### 3.1. Characteristics of Participants

In total, 58 women of reproductive age without any pathological symptoms were recruited from Pemba and 17 women were recruited from Unguja, the most urbanised and touristic island of the Zanzibar archipelago. Details of participant characteristics are reported in Table A1 and Table A2 (Appendix A). Anthropometric parameters are presented in Table 1. The average ages of the selected participants were 30.04 ± 6.31 years and 29.79 ± 7.36 years in Pemba and Unguja, respectively; and the average BMI was significantly high in Unguja compared to Pemba (*p*-value < 0.001). Ten participants from Pemba were obese and twelve were pregnant. Three of them were underweight. The microscopic examination of stools revealed that six participants exclusively carried *Ascaris lumbricoides*, six carried *Trichuris trichiura*, and two participants had a co-infestation (*Ascaris*/*Trichuris*) in Pemba. 

### 3.2. Structure of the Diet of Women of Reproductive Age from Pemba

The analysis of recorded food intake for one week (Figure 1) revealed that women from Pemba (Figure 1A) consume a limited variety of foods: rice, cassava, and bread as sources of carbohydrates. They also consume fish, beans, and vegetables. However, fish consumption was not common among all recruited participants. Other foods consumed by this population are tea, banana, ugali (stiff cassava flour), and porridge. The heatmap of food consumption for women from Pemba revealed that fruits, milk and dairy products, eggs, and poultry are either absent or consumed by very few of the participants. Women from Unguja (Figure 1B) presented a slightly different trend of dietary habits. 

To further our understanding of the dietary trends of the two islands, we performed a comparative analysis of the intake of commonly eaten foods. In Unguja, there was a significantly high consumption of beans, vegetables, and red meat, as well as a relatively high consumption of fish, banana, and rice (Figure 2). Moreover, women from Unguja tended to very frequently consume urojo (also called Zanzibar mix, an energy-dense food characterized by a mixture of many food items) and fruit juice. They also eat significantly less cassava than women from Pemba. Although there was a low consumption of dagaa (an affordable, small, dried fish available on both islands), there was a relatively higher consumption of this food in Pemba than in Unguja. In general, it appears that, despite the availability of a wide variety of foods, women from Pemba and Unguja had a limited consumption of fruits, poultry, and eggs, and almost no consumption of milk and dairy products.

### 3.3. Analysis of Nutrient Intakes for Women in Pemba and Unguja

Following the WHO guidelines for the nutrient intake of African women [28] and the EFSA guideline for micronutrients [29], we calculated that many women achieved the necessary intake of macronutrients, specifically for proteins, fats, and carbohydrates (Table A3; Appendix A). The average protein intake in grams per day is shown in Table 2, with the corresponding recommended nutrient intake value (RNI), and the percentage of women who achieve this in the two islands. The intake of fibre did not reach the recommended 25 g per day in either Pemba or Unguja women (only 37.20% of Pemba and 17% of Unguja women achieved the RNI).

Concerning vitamin intakes (Table 2), fewer than 50% of women from Pemba reached the recommended intake for all vitamins except for vitamin B1 (thiamine), with 69.81% of women reaching the daily RNI in Pemba. The situation was worse for vitamin B2 (riboflavin) and vitamin B5 (pantothenic acid); the percentage of participants reaching the daily RNI was 0% for both vitamins. Only 29.30% and 37.73% of women reached the RNI of vitamin A and folate, respectively. In Unguja, there was a relatively improved condition regarding some vitamins. The recommended intake was reached for vitamin A, vitamin C, vitamin B1, and folate by 76.47%, 52.94%, 94.11%, and 88.23% women, respectively. For vitamin B2, no participant reached the RNI. A comparison between the two islands revealed that there was a relatively high intake of vitamins in Unguja compared to Pemba, except for vitamin B12, the RNI of which was reached by 41% of women on both islands.

### 3.4. Women from Pemba and Unguja Harbour Different Gut Microbiota

The analysis of the 16S rDNA sequencing data revealed that the gut microbiota of Zanzibar women differs according to their location (Pemba and Unguja) (Figure 3), despite being ethnically and culturally close, and there being a regular flow of population between the two islands, as well as genetic connections due to mixed marriages. The alpha diversity (Figure 3A) was statistically different regarding the number of observed OTUs, as well as the Shannon, Evenness pielou, and phylogenetic diversity indexes. The beta diversity (Figure 3B), that is, the qualitative measure of community dissimilarities incorporating phylogenetic relationships between different groups or different ecological environments, also revealed a significant difference regarding the Unweighted UniFrac distance (*p* = 0.032 *).

The composition of the gut microbiota at the phylum and genus levels was different according to location. The differential analysis of taxonomy considering the effect size using the STAMP software showed that, at the phylum level (Figure 3C), Bacteroidetes and Desulfobacterota were significantly more abundant in the gut microbiota of women from Unguja (Zanzibar), while Cyanobacteria and Proteobacteria were significantly more present in the gut microbiota of women from Pemba. 

The analysis of taxonomy at the genus level (Figure 3D) showed that *Blautia*, *Catenibacterium*, *Intestinibacter*, and *Romboutsia* (all members of the Firmicutes), as well as *Collinsella*, *Gastranaerophilales*, *RF39*, and *Sutterellla*, were significantly more present in Pemba. On the other hand, members of the Bacteroidetes phylum, specifically *Prevotella* and *Rikenellaceae_RC9_gut group*, and some members of the Firmicutes phylum, specifically *Dorea*, *UCG-002*, *UCG-003*, were more abundant in Unguja. Despite the high diversity found in Unguja, we noticed that *Escherichia-Shigella*, belonging to the Proteobacteria, was more abundant in women there. 

### 3.5. Correlation Analysis of Foods and Gut Microbiota in Non-Pregnant and Healthy Women from Pemba and Unguja

Spearman’s correlation analysis between the food and gut microbiota of healthy participants revealed consistent correlations in both Pemba (Figure 4A) and Unguja (Figure 4B). We can underline some of these relevant correlations, such as *Clostridium sensu stricto* (potential protective bacteria), which was positively correlated with cassava in Pemba and Unguja; *Faecalibacterium*, encompassing a major actor of human intestinal health; and the species *Faecalibacterium prausnitzii*, consistently and positively correlated with vegetables in both locations. On the other hand, *Escherichia*/*Shigella* (known to favour the *Trichuris* infection) was positively correlated with banana and negatively correlated with cassava in both contexts.

Moreover, we noticed that some correlations were only found in one location. Among correlations that were found only in Unguja (Figure 4B) were *Haemophilus*, which was positively correlated with fish and negatively with chips, and *Treponema*, which was positively correlated with mango and negatively correlated with sembe (corn flour stiff porridge) and vegetables. Likewise, some associations were found only in Pemba (Figure 4B), such as *Methanobrevibacter*, which was positively correlated with sembe, bread, and beans, and negatively correlated with vegetables and dagaa, and *Subdoligranulum*, a potential, protective butyrate-producing bacterium, which was positively correlated with cassava and beans, and negatively correlated with red meat, milk, and fruit juice in Pemba women.

Furthermore, we noticed that there were some bacteria that had different or inconsistent associations with the same food item according to the location. For example, *Clostridia_UCG-014* was positively correlated with sembe (corn flour stiff porridge) in Pemba (Figure 4A), but it showed the opposite correlation with the same food in Unguja (Figure 4B). Likewise, *Faecalibacterium* and sembe were negatively correlated in Pemba, but they showed a positive correlation in Unguja. This suggests that the correlation between gut microbiota and diet may vary according to the host environment or context.

### 3.6. Nutrient and Gut Microbiota Correlations in Non-Pregnant and Healthy Women from Pemba and Unguja

The analysis of the associations between nutrients and gut microbiota in non-pregnant and healthy women showed consistent correlations in both locations and some inconsistent correlations (Figure 4C,D). Firstly, as consistent correlations, *Clostridium_sensu_stricto-1* was positively correlated with vitamin B12 in both islands, indicating that increasing the intake of vitamin B12 may favour the development of this taxon; in the same way, *Faecalibacterium* was consistently positively correlated with calcium and vitamin A; and *UCG*-*002* was consistently positively correlated with copper. Unexpectedly, *Prevotella* was negatively correlated with carbohydrates and fibre in both locations. 

On the other hand, we found correlations that were observed in only one location. In healthy women from Unguja (Figure 4D), *Haemophilus* was positively correlated with magnesium, iron, and vitamin B2; *Clostridium_sensu_stricto-1* was strongly positively correlated with calcium in Unguja. In Pemba (Figure 4C), *Bifidobacterium* had a strong positive correlation with vitamin A. It was also correlated with calcium and magnesium, and negatively correlated with fats. Only in Pemba, we observed that *Subdoligranulum* was positively correlated with fibre, as well as several minerals (Mg, Fe, Zn, Cu, and Mn) and vitamins (B1, B3, and B6). 

Finally, we searched for inconsistent correlations and noticed that some associations vary depending on the location. For example, *Clostridium_sensu_stricto-1* was positively correlated with vitamin E in Pemba but had a negative correlation in Unguja; similarly, *UCG-002* and vitamin B1 were positively correlated in Pemba (Figure 4C) but negatively correlated in Unguja (Figure 4D). 

### 3.7. Correlation Analysis of Foods and Gut Microbes in Helminth-Infected Participants

Since it has been demonstrated that helminth infection has an impact on the composition of the gut microbiota, we conducted a test to determine whether the food/nutrient–microbiota correlations were altered in the guts of infected women. Figure 5 reveals new consistent associations between bacteria and foods in *Ascaris* (Figure 5A) and *Trichuris* (Figure 5B) infections. Amidst those correlations, we noticed that *Prevotella*, the most abundant genus of the African population, was positively correlated with banana and negatively associated with vegetables in both types of helminth conditions. *Eubacterium coprostanoligenes_group* was invariably positively correlated with cassava and negatively with buns (bread rolls). *Bacteroides*, one the most abundant genus of Bacteroidetes, was consistently positively correlated with porridge and vermicelli. *Escherichia/Shigella*, an undesirable genus since it is generally associated with *Trichuris* infection, was consistently and negatively correlated with banana and dagaa in both *Ascaris* and *Trichuris* infection conditions. 

We also looked for correlations found only in one or the other helminth condition. In *Ascaris* infection, the *Lachnospiraceae-NK4136* group was strongly and positively correlated with sembe (corn flour stiff porridge) and negatively correlated with ugali; *Subdoligranulum* was positively correlated with fish and cassava in *Ascaris* infection (Figure 5A). In the context of *Trichuris* infection (Figure 5B), we noticed that *Bifidobacterium*, a well-characterised genus in children and healthy gut microbiota, was positively correlated with beans, vegetables, porridge, and oranges, and negatively correlated with buns and ubuyu. Other relevant associations were as follows: *Alloprevotella* was positively correlated with dagaa, bananas, and oranges, and negatively correlated with breads and ground nuts; and *Subdoligranulum* was positively correlated with beans, porridge, and oranges in the same context.

### 3.8. Correlation Analysis of Nutrients and Gut Microbes in Helminth-Infected Participants

Finally, we analysed Spearman’s correlations of nutrients and microbiota in *Ascaris*- and *Trichuris*-infected participants (Figure 5C,D) and observed new and consistent correlations. Here, *Faecalibacterium*, which was positively correlated with calcium in healthy participants, maintained this correlation and showed a new correlation with vitamin B12 under both helminth infection conditions. The *Eubacterium coprostanoligenes group* was positively and consistently correlated with fibre, vitamin C, vitamin B6, and folate in both *Ascaris*- and *Trichuris*-infected participants. We also observed that *Bacteroides* was positively and consistently correlated with iron under both conditions. Interestingly, *Escherichia-Shigella* was negatively and invariably correlated with vitamin B12 and vitamin B2, indicating that a supplementation of these vitamins may be useful for limiting the growth of *Escherichia-Shigella*, which is usually associated with *Trichuris* infection. We also noticed that *Subdoligranulum* presented more positive correlations with minerals and B vitamins, suggesting the affinity of this genus with healthy diets rich in micronutrients. 

On the other hand, some correlations were found either only in *Ascaris* or only in *Trichuris* infections. As examples of correlations appearing only in *Ascaris* infection (Figure 5C), *Roseburia* was positively correlated with vitamin E and negatively correlated with phosphorus; and *Coprococcus* was positively correlated with vitamin E, vitamin C, and folate, and negatively correlated with protein, carbohydrates, and phosphorus.

Similarly, some correlations were observed only in the context of *Trichuris* infection, as follows: *Clostridium sensu_stricto-1* (Figure 5D) was positively and strongly correlated with vitamin B12 and negatively correlated with iron; *Bifidobacterium*, in the same context, was positively correlated with folate and vitamin A, and negatively correlated with fat and protein; *Alloprevotella* appeared to be positively correlated with calcium, vitamin B2, and vitamin B12, and negatively correlated with vitamin E. 

An unexpected finding was that the *Prevotella* genus, an important member of the core microbiota of the African population, showed unusual patterns of correlations. It showed strong positive correlations with protein, fat, carbohydrates, and vitamin B12 in *Trichuris* infection (Figure 5D). 

## 4. Discussion

The analysis of the diet of Zanzibar women revealed that the food they consume is not diverse enough, despite the wide availability of several varieties of locally produced foods, such as fruits and vegetables. According to the World Health Organization, eating plenty of vegetables and fruits is recommended since they are important sources of vitamins, minerals, dietary fibre, plant protein, and antioxidants; people with diets rich in vegetables and fruit have a significantly lower risk of obesity and other diseases. Moreover, the low or absent consumption of milk products should be considered not favourable for the health of this population, since milk and dairy products are recognised as nutrient-dense foods, supplying energy and high-quality proteins with a range of essential micronutrients [30]. Milk and dairy products can play important roles in human nutrition in developing countries, where the diets of poor people frequently lack diversity and the consumption of foods from animal sources may be limited [31].

Since their diet is not diverse enough, most women of Pemba and Unguja reach the recommended nutrient intakes (RNI) for macronutrients but have serious deficiencies in vitamins and minerals. Participants had significant deficiencies regarding the recommended intake of B vitamins (B2, B3, B5, B6, and B12). The RNI of vitamin A was not reached. The importance of B vitamins is clear; they are precursors of essential cofactors used in many metabolic pathways. This makes them essential for the development of the host and gut microbiota. Mammalian hosts are not able to produce vitamin B de novo. Therefore, they are dependent on the intake from diet and gut microbiota activity [32]. In research using in silico analysis, Rodionov et al. showed that 20–30% of gut bacteria lack the capacity to produce essential B vitamins [33]. This deficiency may therefore affect not only host health in general but also the gut microbiota.

The deficiency in minerals is worrying, specifically calcium, iron, and potassium. Only 13.20% and 5.88% of women reached the daily RNI of iron in Pemba and Unguja, respectively. This condition has been a concern in Zanzibar since 2018, when the prevalence of anaemia among women ranged from 37.5% in Stone Town to 49.2% in Pemba South [34]. It has been demonstrated that iron deficiency impacts not only the host, but also microbiota efficiency [35]. A previous study conducted on rats showed that iron-deficient rats had considerably lower concentrations of butyrate and propionate [36]. The deficiency in calcium intake is also critical, since calcium is essential for the activities of muscles, nerves, and blood vessels, the release of hormones that affect many other functions in the human body, and for the gut microbiota function. Chaplin et al. found that calcium acts in a prebiotic manner to influence the gut microbiota in mice [37]. Calcium-fed animals exhibited increased levels of *Bifidobacterium* spp., *Bacteroides*, and *Prevotella*. Whisner et al. found that the daily consumption of galacto-oligosaccharides increases calcium absorption, which may be mediated by the gut microbiota, specifically by *Bifidobacterium* [38]. 

The alpha and beta diversity of the gut microbiota of women from Pemba and Unguja differ according to location even though they are ethnically close. This agrees with the study by Mehta et al., which demonstrated that residence location was associated with differences in gut microbiome composition in Indian adults [39]. This difference can be explained by the possible influence of the environment [40], more rural in one case, or the difference in their diets [13]. We found that women from Unguja (Figure 1B) have slightly different dietary habits with some specifications. They have a significantly high consumption of beans, vegetables, and red meat, and a relatively high consumption of fish, banana, and rice (Figure 2). Moreover, they tend to consume urojo very frequently (an energy-dense food made from a mixture of many food items) and fruit juice compared to women from Pemba. They also eat significantly less cassava compared to women from Pemba. The relative abundance analysis, which revealed that Bacteroidetes were more abundant in Unguja, can be explained by their diet being more varied than that of Pemba participants. This finding also aligns with Martinez et al., 2017 study, which concluded that urbanization correlates with distinct gut microbiota composition compared to rural populations [41]. At the genus level, the dominant taxa in Pemba compared to Unguja are mostly members of the Firmicutes phylum (*Blautia*, *Romboutsia*, *Intestinibacter*, and *Catenibacter*), while in Unguja, the most dominant taxa are either from the Bacteroidetes (*Prevotella* and Rikenellaceae_RC9_group) or the Firmicutes (*Dorea*, *UCG-002*, and *UCG-003*). Our findings agree with a study conducted by Elsherbiny et al., in which the abundance of many genera was strongly dependent on geographical location [42]. 

Spearman’s correlation analysis showed that some bacterial taxa were consistently associated with the same food or nutrients in healthy women from both locations or in both types of helminth infection. Many studies demonstrated that diet influences the composition [43] and function [44] of the gut microbiota. In our study, *Clostridium sensu stricto*-1, which was previously negatively associated with *Trichuris* infection in the studies of Rosa et al. [45] and Cooper et al. [46], was consistently correlated with cassava in both Pemba and Unguja. It was also consistently and positively correlated with vitamin B12 for both islands. This indicates that the increased consumption of cassava, rich in dietary fibre, can favour the development of this taxon. Additionally, vitamin B12 may also increase the abundance of *Clostridium sensu stricto*-1. Vitamin B12 (Cobalamin) uptake by *Bacteroides thetaiomicron* is observed to be able to limit shiga toxins, acting as an immunomodulator to promote cellular immunity [32]. Likewise, *Faecalibacterium*, recognised as a producer of short-chain fatty acids (SCFAs), was consistently and positively correlated with vegetables, vitamin A, and calcium in healthy women from both locations but conserved only the positive correlation with calcium in both *Ascaris* and *Trichuris* infection, suggesting the strong affinity of this genus with calcium. The prebiotic properties of calcium have been recognized in humans and mice [47]. Importantly, *Escherichia-Shigella*, which is usually associated with gut problems and with *Trichuris* infection [22,23,45], was invariably and negatively associated with cassava at both locations, indicating that the increased consumption of this fibre may limit the proliferation of this taxon and consequently reduce infections. This also justifies why we found a high abundance of *Escherichia-Shigella* in the microbiota of women from Unguja, who consume significantly less cassava compared to women from Pemba. Moreover, we noticed that *Escherichia-Shigella* was also negatively correlated with vitamin B2 and vitamin B12 in both *Ascaris* and *Trichuris* infections. A preliminary study showed that the supplementation of riboflavin (vitamin B2) increased *F. prausnitzii* and concomitantly reduced *Escherichia-Shigella* in a small group of adults [48]. Another relevant observation was the positive correlation of *Eubacterium coprostanoligens* with cassava and its negative correlation with buns in healthy women. *Eubacterium coprostanoligens* was found to be part of the enterotype that favourably responds to Albendazole- and Ivermectin-based treatments [22]. This means that an increase in the abundance of *Eubacterium* may be helpful in fighting against helminths for this population. Likewise, *Bacteroides,* which is usually not associated with *Trichuris* infection in humans [24,49], was positively correlated with porridge and iron in both infections. Our findings here are in agreement with a study on Korean adolescents, which also found that *Bacteroides* was associated with iron, especially plant iron [14]. *Subdoligranulum*, which might contribute to resistance to helminth infection in self-clearing individuals [50,51], was positively correlated with cassava, beans, porridge, and vermicelli. It was also positively and strongly correlated with magnesium, zinc, copper, vitamin B1, vitamin B3, and folate. This suggests that *Subdoligranulum* abundance is an indicator of a balanced micronutrient diet. 

Moreover, the results of Spearman’s correlation analysis also revealed that helminths may affect the relationships between diet and gut microbiota in a helminth-species-dependant manner. This was illustrated by correlations that were only found in either *Ascaris* or *Trichuris* infections. It has already been demonstrated that helminth infection influences the response of the gut microbiota to dietary interventions [52]. Our analysis revealed that this influence may differ according to the helminth species. 

Finally, the *Prevotella* genus, an important member of the gut microbiota associated with African diets [53], which predominantly contain carbohydrates [54], showed a consistent positive correlation with bananas and a negative correlation with vegetables. Unexpectedly, our analysis revealed that the association between diet and this genus is not as simple as originally thought. The *Prevotella* genus, in our investigation, was consistently negatively correlated with carbohydrates and fibre in both locations. A study conducted by De Filippis et al. [16] revealed that some *Prevotella* oligotypes were significantly associated with plant-based diets while others were associated with animal-based nutrients. This highlights the importance of considering the sub-genus level when studying the diet–microbiota interactions. 

Our study revealed that, although diet/nutrient and gut microbiota associations can vary based on several factors such as the host health and environment, associations between bacteria and food/nutrients that are consistent in healthy women from both locations meant that the increased consumption of specific foods and nutrients can modulate the gut microbiota and potentially improve host health by either increasing good bacteria or limiting the proliferation of unwanted bacteria that promote helminth infections.

However, this study had some limitations, such as the relatively limited number of participants and the fact that the dietary data were derived from self-reported food intake. Correlations observed between food and genera are insufficient since we know that, in the same genus, different species or strains may present different metabolic responses to the same foods. Moreover, this study analysed the association, but not the causation relationships, between food/nutrients and microbiota. Furthermore, the differences in microbiota composition according to the origin of participants, as well as the differences in patterns of association between diet and microbiota depending on location, limit the potential to globalise our findings. 

## 5. Conclusions

This study used a combination of self-reported food intake from women of reproductive age and the 16S rDNA sequencing approach to evaluate diets, nutrient intakes, the gut microbiota composition, and its correlations with food or nutrients for women living on the two main islands of the Zanzibar archipelago, where helminth infections are endemic. We found that, despite the availability of many locally produced foods, the diet of women of reproductive age in Zanzibar was not diverse enough. Consequently, most participants had vitamin and mineral deficiencies, particularly for B vitamins, vitamin A, and calcium. This sufficiently demonstrated the need for the nutritional education of the population on the composition of a balanced diet. The analysis of correlations between food/nutrients and gut microbiota revealed consistent associations between food or nutrients and gut bacteria. Associations between specific food, such as cassava, beans, dagaa, fish, and porridge, lead us to presume that increasing the consumption of these food items with the supplementation of B vitamins, vitamin A, iron, and calcium can support the growth of beneficial bacteria or limit the proliferation of undesirable bacteria, and thus make the participants more resistant to helminth infections. Of course, we cannot ignore the socioeconomical conditions of the populations and the difficulties to obtain micronutrient supplementation. However, we had evidence that the specific food that we recommend after this investigation is sustainable for the population living in villages in both islands. 

A nutritional intervention in a large cohort with the same population is required to further support our understanding and to confirm results obtained in this cross-sectional study.

## Figures and Tables

**Figure 1 nutrients-16-01266-f001:**
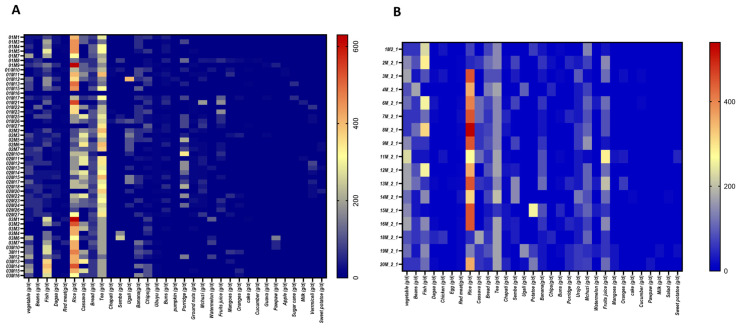
Heatmap of the foods consumed by women in the Zanzibar archipelago. (**A**): Pemba cases. (**B**): Unguja cases. The food intake is evaluated in terms of quantity of grams eaten per day.

**Figure 2 nutrients-16-01266-f002:**
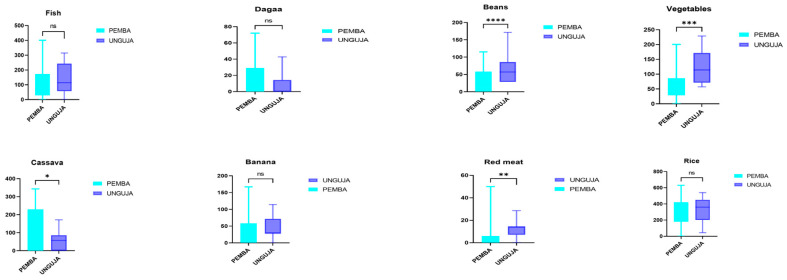
Comparative analysis of the average dietary intakes (in grams) of the most consumed foods in Zanzibar. An unpaired t-test was used. Fish (*p*-value = 0.331), dagaa (*p*-value = 0.096), beans (*p*-value < 0.0001), vegetables (*p*-value =0.0001), cassava (*p*-value = 0.015), banana (*p*-value = 0.418), red meat (*p*-value = 0.004), rice (*p*-value = 0.448). A *p*-value < 0.05 was considered as significant. * means *p*-value < 0.05; ** means *p*-value < 0.01; *** means *p*-value < 0.001; **** means *p*-value < 0.0001; ns means *p*-value > 0.05.

**Figure 3 nutrients-16-01266-f003:**
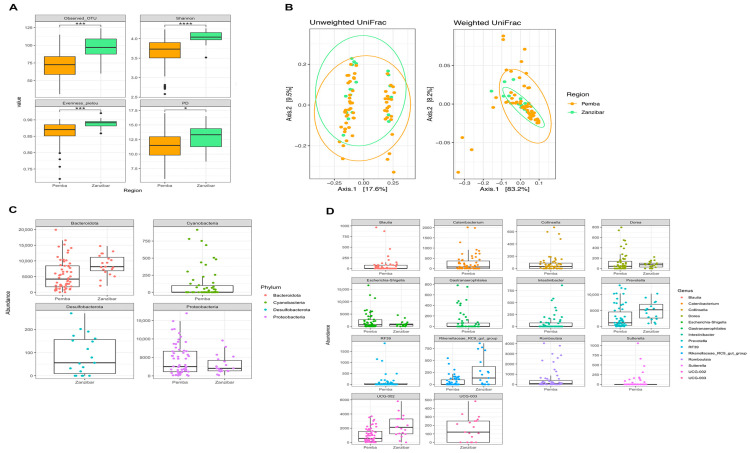
Diversity and taxonomy analyses of the gut microbiota of women of reproductive age from the Zanzibar archipelago according to location. (**A**) Alpha diversities. (**B**) Beta diversities (*p*-value = 0.032 * for the Unweighted UniFrac, *p*-value = 0.776 for the Weighted UniFrac). (**C**) Taxa that significantly changed at the phylum level. (**D**) Taxa that significantly changed at the genus level. In this figure, Zanzibar refers to Unguja Island. If only one bar is shown in the plot, it indicates that the values for the other variables are all zero. * means *p* ≤ 0.05; *** means *p* ≤ 0.001; **** means *p* ≤ 0.0001.

**Figure 4 nutrients-16-01266-f004:**
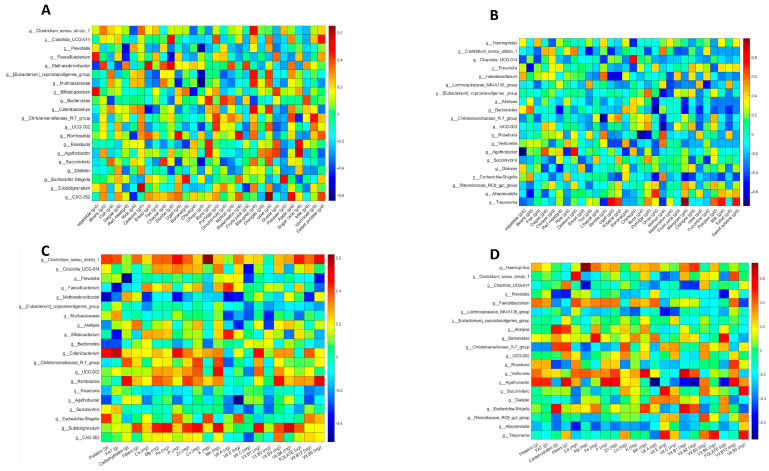
Heatmaps showing food–gut microbiota correlations in non-pregnant and healthy women in Pemba (**A**) and Unguja (**B**). Heatmaps of nutrients and gut microbiota correlations in Pemba (**C**) and Unguja women (**D**).

**Figure 5 nutrients-16-01266-f005:**
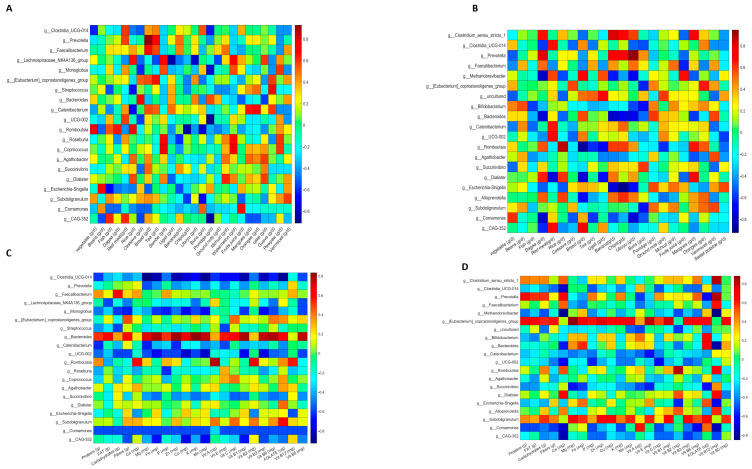
Heatmaps of the correlations between food or nutrients and gut microbiota in infected participants. (**A**) Food–microbiota correlations in *Ascaris lumbricoides*-infected participants. (**B**) Food–microbiota correlations in *Trichuris trichiura*-infected participants. (**C**) Nutrient–microbiota correlations in *Ascaris lumbricoides*-infected participants. (**D**) Nutrient–microbiota correlations in *Trichuris trichiura*-infected participants.

**Table 1 nutrients-16-01266-t001:** Anthropometric characteristics of selected participants from Pemba and Unguja.

	Pemba (Mean ± Std)	Unguja (Mean ± Std)
Age (years)	30.04 ± 6.31	29.79 ± 7.36
Weight (kg)	59.29 ± 13.30	59.8 ± 11.19
Height (cm)	155.89 ± 5.95	138.52 ± 16.46
BMI (kg/m^2^)	24.34 ± 4.88	32.22 ± 9.13

**Table 2 nutrients-16-01266-t002:** Macro- and micronutrient intakes of women from Zanzibar archipelago. Recommended micronutrients are established according to the WHO guidelines for the nutrient profiles of African women that define the ranges for each nutrient intake per day. Macronutrients are measured in grams(g) while micronutrients are measured in milligrams (mg) or micrograms (μg).

Nutrient	Intake (Pemba)	Intake (Unguja)	RNI	% of Women Reaching the RNI (Pemba)	% of Women Reaching the RNI (Unguja)
Protein (g)	67.59 ± 21.19	76.87 ± 19.46	50–75	84.61	88.24
Fat (g)	72.59 ± 26.95	71.16 ± 18.68	33.33–66.66	96.15	88.24
Carbohydrates (g)	290.84 ± 52.85	274 ± 60.23	275–375	75	58.82
Fibre (g)	18.94 ± 7.82	21.60 ± 5.29	25	37.20	17
Vitamin A (μg)	499.89 ± 467.43	1124.12 ± 449.41	650	28.30	76.47
Vitamin E (mg)	4.4 ± 1.77	6.91 ± 4.40	11	0.0	11.76
Vitamin C (mg)	85.23 ± 44.28	91.54 ± 41.30	95	43.39	52.94
Vitamin B1 (Thiamine) (mg)	0.91 ± 0.23	1.08 ± 0.21	0.83	69.81	94.11
Vitamin B2 (riboflavin) (mg)	0.78 ± 0.22	1.04 ± 0.23	1.6	0.0	0.0
Vitamin B3 (Niacin) (μg)	8.34 ± 3.13	10.17 ± 2.69	14	7.54	11.76
Vitamin B6 (pyridoxine) (mg)	1.18 ± 0.36	1.55 ± 0.47	1.6	11.32	29.41
Folate (μg)	264.79 ± 109.07	428.21 ± 95.90	330	37.73	88.23
Vitamin B12 (Cobalamin) (μg)	2.76 ± 2.53	2.55 ± 1.61	2.4	41.50	41.17
Vitamin B5 (pantothenic acid) (mg)	2.51 ± 0.83	4.25 ± 2.75	5	0.0	17.64
Calcium (mg)	512.34 ± 360.05	599.45± 219.83	950	13.20	11.78
Magnesium (mg)	323.51 ± 112.05	367.85 ± 68.07	300	49.05	94.11
Iron (mg)	10.81 ± 5.01	12.25 ± 2.57	16	13.20	5.88
Phosphorus (mg)	1258 ± 451.03	1253.49 ± 336.45	550	94	100
Zinc (mg)	7.12 ± 2.26	7.64 ± 1.39	7.5	41.50	47.05
Copper (mg)	1.46 ± 0.60	1.49 ± 0.39	1.3	50.94	70.64
Potassium (mg)	2574.25 ± 825	2912.70 ± 545.41	3500	15	17.64
Manganese (mg)	5.15 ± 2.12	4.58 ± 1.11	3.0	88.67	94.11

Mineral intakes were higher than vitamin intakes. The average intake reached the RNI for magnesium, phosphorus, copper, and manganese in both islands. However, looking at the proportion of participants reaching the RNI, many women did not take enough calcium (13.20% in Pemba and 11.78 in Unguja), iron (13.20% in Pemba and 5.88% in Unguja), or potassium (15% in Pemba and 17.64% in Unguja).

## Data Availability

The original data presented in the study are openly available from the publication of this article in the NCBI Sequence Read Archive database (SRA accession number: SRP495566, BioProject accession number: PRJNA1088637).

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
