# Peer review of "Association between Food or Nutrients and Gut Microbiota in Healthy and Helminth-Infected Women of Reproductive Age from Zanzibar, Tanzania"

_nutrients, 2024, doi:10.3390/nu16091266_

Round 1
Reviewer 1 Report
Comments and Suggestions for Authors
Please see the attached report.

Reviewer 2 Report
Comments and Suggestions for Authors
The main goal of the paper presented was to explore the correlations between diet and gut microbiota in both healthy and helminth-infected women of reproductive age in Pemba and Unguja, the two main islands of Zanzibar, where child stunting and malnutrition are monitored by the WHO. The final objective was to identify correlations between specific foods or nutrients that may promote the growth of beneficial bacteria or limit the proliferation of harmful bacteria.
Relevance of the work:
The work presented by the authors is of particular interest because it fills a substantial knowledge gap concerning the interplay among diet, gut microbiota, and helminth infections, particularly in regions where these infections are endemic.
By investigating the composition of gut microbiota in reproductive-aged women in Zanzibar, the study sheds light on location-specific variations and uncovers dietary inadequacies, offering valuable insights into regional health conditions.
Furthermore, by establishing correlations between the abundance of specific bacteria and deficiencies in diet components (namely vitamins, iron and calcium), authors highlight potential mechanisms through which diet may influence susceptibility to helminth infections. These findings suggest the possibility of targeted dietary interventions to enhance gut microbiota resilience against helminth infections, thereby presenting a potentially efficacious approach for preventing and managing these diseases in endemic regions.
General considerations:
This work presents significant scientific relevance, being well structured and designed to address important issues in the field of intestinal microbiota, diet, and helminth infections in vulnerable populations. However, it is important to note that the number of samples used for microbiota analysis is limited, which may restrict the generalization of the results.
Additionally, it is necessary to note that the quality of the provided images is very poor, which could compromise the understanding of the presented data. It was really difficult to analyse the images with the quality provided. Therefore, these images must be improved before the work is considered for publication.
Another point to be highlighted is the need for a detailed review of the bibliographic references to ensure uniformity and accuracy.
Overall, this work has significant merits but requires some improvements before being considered for publication.
Suggestions for authors:
- Line 206: The units of BMI in table 1 are missing, please correct.
- Line 206: In table 1 the authors wrote mean +/- sdt, is this correct? The abbreviation for standard deviation usually is SD or Std.
- Line 261: In table 2 the units for the intake are missing (column 1 and 2).
- Line 415: In the sentence “despite there several varieties of locally” do you really want to say “there”?
- Line 487: When authors talk about Escherichia-Shigella, I think is important to give some context regarding these bacteria, namely saying that they are associated with gut related problems.
- Line 588: Why do you have a different section for the references used during the analysis?
- Line 602: Appendix B – again attention to the units that are missing in the table.
- Line 664: Reference 28: This is a book chapter, right? The reference is incomplete. Please correct.
- Line 698: Reference 45 – I can´t find this article. Maybe the reference is incomplete.
Comments on the Quality of English Language
Minor editing of English language required. Please revise the units and abbreviations used.
Reviewer 3 Report
Comments and Suggestions for Authors
Current study investigated the interactions among diet, microbiota, and helminth infection.
Abstract:
Line 28: “specific food and nutrients”, please specify what specific food and nutrients that the authors are indicating.
Line 29: “potentially making it more resilient to helminth infections in endemic areas”. It speculates too much. I would recommend to remove this sentence. Or if you would need additional data/discussion to support the statement.
M&M
How did the authors decide on the number of participants? In total, the study included 75 participants, while later on (at line 146-149) the number of participants decreased sharply (18 Pemba, 13 WRA). Moreover, how did you select (as in line 151) the infected subjects? Were they from the 75 participants? Follow diagram is urgently needed to give info about your recruited subjects.
Line:113-114, it should be in your statistical analysis section.
Line 183-184, the study used both Silva and Greengenes databases, did the result the same? What are the differences?
Line 189-191, this info should be in section 2.6
Section 2.7. Too little info was added for the statistical analysis. Please provide more details.
Result and discussion:
58 women from Pemba, 17 from Unguja. Again, how this number was decided?
Table 1: any significant differences between women from these two places?
Figure 1: what is your input for Figure 1? Presence or not? Or intake intensity/frequency? Please add this information to Figure legend.
Figure 2, what is the unit for figure 2 y axis?
Table 2, the second and third column, what are the unit?
Figure 3, figure legend of 3D is way too small.
What are the differences in the intestinal microbiota of infected and healthy subjects?
Comments on the Quality of English Language
The authors of adequate skills in English writing.
Round 2
Reviewer 3 Report
Comments and Suggestions for Authors
No additional comments. My questions have been addressed.
Author Response
We express our gratitude to all reviewers for their contributions.